# REGULARIZATION SHORTCOMINGS FOR CONTINUAL LEARNING

## ABSTRACT

In most machine learning algorithms, training data is assumed to be independent and identically distributed (iid). When it is not the case, the algorithms performances are challenged, leading to the famous phenomenon of *catastrophic forgetting*. Algorithms dealing with it are gathered in the "Continual Learning" research field. In this paper, we study the *regularization* based approaches to continual learning and show that those approaches can not learn to discriminate classes from different tasks in an elemental continual benchmark, the class-incremental setting. We make theoretical reasoning to prove this shortcoming and illustrate it with experiments. Moreover, we show that it can have some important consequences on multi-tasks reinforcement learning or in pre-trained models used for continual learning. We believe this paper to be the first to propose a theoretical description of regularization shortcomings for continual learning.

## 1 INTRODUCTION

Continual Learning is a sub-field of machine learning dealing with non-iid (identically and independently distributed) data French (1999); Lesort et al. (2019c). Its goal is to learn the global optima to an optimization problem where the data distribution changes through time. This is typically the case in databases that get regularly augmented with new data or when data is streamed to the algorithms with limited storage possibilities.

Continual learning (CL) looks for alternative methods to the iid training to avoid the complete retraining with all data each time new data is available. CL algorithms propose different memory storage approaches to collect information from past learning experiences and learning algorithms to continue to learn with this memory and new data.

In this paper, we propose to study the class-incremental setting with regularization based methods. The class-incremental setting consists of learning sets of classes incrementally. Each task is composed of new classes. As the training ends, the model should classify data from all classes correctly. Without task labels for inferences, the model needs to both learn the discrimination of intra-task classes and the trans-task classes discrimination (i.e. distinctions between classes from different tasks). On the contrary, if the task label is available for inferences, only the discrimination of intra-task classes needs to be learned. The discrimination upon different tasks is given by the task label. Learning without access to task labels at test time is then much more complex since it needs to discriminate data that are not available at the same time in the data stream.

In such setting, we would like to demonstrate that regularization does not help to learn the discrimination between tasks. For example, if a first task is to discriminate white cats vs black cats and the second is the same with dogs, a regularization based method does not provide the learning criteria to learn features to distinguish white dogs from white cats.

We consider as regularization methods, those who aim at protecting important weights learned from past tasks without using a buffer of old data or any replay process. Those methods are widely used for continual learning Kirkpatrick et al. (2017); Zenke et al. (2017); Ritter et al. (2018); Schwarz et al. (2018). In this paper, we show that in the classical setting of class-incremental tasks, this approach has theoretical limitations and can not be used alone. Indeed, it does not provide any learning criteria to distinguish classes from different tasks. Therefore, in practice, regularization algorithms need external information to make an inference in class-incremental settings. It is provided by the task

label at test time. However, relying on the task label to make inferences is an important limitation for algorithms' autonomy, i.e. its capacity to run without external information, in most application scenarios.

We believe this paper presents important results for a better understanding of CL which will help practitioners to choose the appropriate approach for practical settings.

## 2 RELATED WORKS

In continual learning, algorithms protect knowledge from catastrophic forgetting French (1999) by saving them into a memory. The memory should be able to incorporate new knowledge and protect existing knowledge from modification. In continual learning, we distinguish four types of memorization categories *dynamic architecture* Rusu et al. (2016); Li & Hoiem (2017), *rehearsal* Chaudhry et al. (2019); Aljundi et al. (2019); Belouadah & Popescu (2018); Wu et al. (2019); Hou et al. (2019); Caccia et al. (2019), *generative replay* Shin et al. (2017); Lesort et al. (2019a); Wu et al. (2018) and *regularization* Kirkpatrick et al. (2017); Zenke et al. (2017); Ritter et al. (2018); Schwarz et al. (2018).

In this paper, we are interested in the capacity of making inferences without task labels at test time (test task label). The task label $t$ (typically a simple integer) is an abstract representation built to help continual algorithms to learn. It is designed to index the current task and notify if the task changes Lesort et al. (2019c). *Dynamic architecture* is a well-known method that needs the task label at test time for an inference. Indeed, since the inference path is different for different tasks, the task test label is needed to use the right path through the neural network Rusu et al. (2016); Li & Hoiem (2017). Rehearsal and Generative Replay methods generally need the task label at training time but not for inferences Lesort et al. (2019a;b). Finally, Regularization methods are often assumed as methods that need task labels only at training time. In this article, we show that in class-incremental settings, it is also necessary at test time.

Test task labels have been used in many continual learning approaches, in particular in those referred to as "multi-headed" Lange et al. (2019). However, the need for task labels for inferences makes algorithms unable to make autonomous predictions and therefore we believe that this requirement is not in the spirit of continual learning. Continual learning is about creating autonomous algorithms that can learn in dynamic environments Lesort et al. (2019c).

## 3 REGULARIZATION APPROACH

In this section, we present the formalism we use and we present the class-incremental learning problem with a regularization based approach.

### 3.1 FORMALISM

In this paper, we assume that the data stream is composed of $N$ disjoint tasks learned sequentially one by one (with $N >= 2$). Task $t$ is noted $T_t$ and $\mathbb{D}_t$ is the associated dataset. The task label $t$ is a simple integer indicating the task index. We refer to the full sequence of tasks as the continuum, noted $C_N$. The dataset combining all data until task $t$ is noted $\mathbb{C}_t$. While learning task $T_t$, the algorithm has access to data from $\mathbb{D}_t$ only.

We study a disjoint set of classification tasks where classes of each task only appear in this task and never again. We assume at least two classes per task (otherwise a classifier cannot learn).

Let $f$ be a function parametrized by $\boldsymbol{\theta}$ that implement the neural network's model. At each task $t$ the model learn an optimal set of parameters $\boldsymbol{\theta}_t^*$ optimizing the task loss $\ell_{\mathbb{D}_t}(\cdot)$. Since we are in a continual learning setting, $\boldsymbol{\theta}_t^*$ should also be an optima for all tasks $T_{t'}$, $\forall t' \in [\![0, t]\!]$.

We consider the class-incremental setting with no test task label. It means that an optima $\boldsymbol{\theta}_1^*$ for $T_1$ is a set of parameters which at test time will, for any data point $x$ from $\mathbb{D}_0 \cup \mathbb{D}_1$, classify correctly without knowing if $x$ comes from $T_0$ or $T_1$. Therefore, in our continual learning setting, the loss to optimize when learning a given task $t$ is augmented with a remembering loss:

$$\ell_{\mathbb{C}_t}(f(\boldsymbol{x}; \boldsymbol{\theta}), y) = \ell_{\mathbb{D}_t}(f(\boldsymbol{x}; \boldsymbol{\theta}), y) + \lambda \Omega(C_{t-1}) \tag{1}$$

where $\ell_{\mathbb{C}_t}(.)$ is the continual loss, $\ell_{\mathbb{D}_t}(.)$ is the current task loss, $\Omega(C_{t-1})$ is the remembering loss with $C_{t-1}$ represents past tasks, $\lambda$ is the importance parameter.

## 3.2 PROBLEM

In continual learning, the regularization approach is to define $\Omega(\cdot)$ as a regularization term to maintain knowledge from $C_{t-1}$ in the parameters $\boldsymbol{\theta}$ such as while learning a new task $T_t$, $f(x; \boldsymbol{\theta}_{t-1}^*) \approx f(x; \boldsymbol{\theta})$, $\forall \boldsymbol{x} \in \mathbb{C}_{t-1}$. In other words, it aims to keep $\ell_{\mathbb{C}_{t-1}}(f(x; \boldsymbol{\theta}), y)$ low $\forall \boldsymbol{x} \in \mathbb{C}_{t-1}$ while learning $T_t$.

The regularization term $\Omega_{t-1}$ act as a memory of $\boldsymbol{\theta}_{t-1}^*$. This memory term depends on the learned parameters $\boldsymbol{\theta}_{t-1}^*$, on $\ell_{\mathbb{C}_{t-1}}$ the loss computed on $T_{t-1}$ and the current parameters $\boldsymbol{\theta}$. $\Omega_{t-1}$ memorizes the optimal state of the model at $T_{t-1}$ and generally the importance of each parameter with regard to the loss $\ell_{\mathbb{C}_{t-1}}$. We note $\Omega_{\mathbb{C}_{t-1}}$ the regularization term memorizing past tasks optimal parameters.

When learning the task $T_t$, the loss to optimize is then:

$$\ell_{\mathbb{C}_t}(f(x; \boldsymbol{\theta}), y) = \ell_{\mathbb{D}_t}(f(x; \boldsymbol{\theta}), y) + \lambda \Omega_{\mathbb{C}_{t-1}}(\boldsymbol{\theta}_{t-1}^*, \ell_{\mathbb{C}_{t-1}}, \boldsymbol{\theta}) \tag{2}$$

Eq. 2 is similar to eq. 1 but in this case the function $\Omega(\cdot)$ is a regularization term depending on past optimal parameters $\boldsymbol{\theta}_{t-1}^*$, loss on previous tasks $\ell_{\mathbb{C}_{t-1}}$ and the vector of current model parameters $\boldsymbol{\theta}$ only. It could be, for example, a matrix pondering weights importance in previous tasks Kirkpatrick et al. (2017); Ritter et al. (2018); Zenke et al. (2017).

# 4 PROPOSITIONS

In this section, we present the proposition concerning the shortcomings of regularization methods in class-incremental settings. We first present definitions and lemmas to prepare for the proposition.

## 4.1 PRELIMINARY DEFINITION / LEMMA

**Definition 1.** *Linear separability*
*Let $S$ and $S'$ be two sets of points in an n-dimensional Euclidean space. $S$ and $S'$ are linearly separable if there exists $n + 1$ real numbers $\omega_1, \omega_2, ..., \omega_n, k$ such that $\forall \boldsymbol{x} \in S$, $\sum_{i=1}^{n} \omega_i x_i > k$ and $\forall \boldsymbol{x} \in S'$, $\sum_{i=1}^{n} \omega_i x_i < k$*

where $x_i$ the $i$th component of x. This means that two classes are linearly separable in an embedded space if there exists a hyperplane separating both classes of data points.

This property can also be written, $\forall \boldsymbol{x} \in S$ and $\forall \boldsymbol{x'} \in S'$, $(\boldsymbol{q} \cdot \boldsymbol{x} + q_0) \cdot (\boldsymbol{q} \cdot \boldsymbol{x'} + q_0) < 0$. With $\boldsymbol{q} = [\omega_1, \omega_2, ..., \omega_n]$ and $q_0 = -k$ respectively the normal vector and position vector of a hyperplane $\mathcal{Q}$. In the case of learning a binary classification with linear model, the model is a hyperplane separating two dataset. As soon as this equation can be solved, then it is possible to define a function $f(\boldsymbol{x}, \theta)$ and a loss $\ell(.)$ to learn a hyperplane that will separate $S$ and $S'$ perfectly.

**Definition 2.** *Interferences*
*In machine learning, interferences are conflicts between two (or more) objective functions leading to prediction errors. There are interferences when optimizing one objective function degrades the optimization of, at least, another one.*

As such, optimizing one of the objective function increases the error on the other one. In continual learning, interferences happen often after a drift in the data distribution. The loss on previous data is increased with the optimization of the loss for the new data leading to interferences and catastrophic forgetting.

**Lemma 4.1.** $\forall(S, S')$ *bounded set of discrete points in $R^n$ and linearly separable by a hyperplane $\mathcal{Q}$. For any algorithm, it is impossible to assess $\mathcal{Q}$ as a separation hyperplane without access to $S'$ set.*

The proof is in appendix B, but in an insightful way, for any bounded set of points $S$, there is a infinite number of linearly separable set of points. Thus, there exists an infinite number of potential

separating hyperplanes. If the second set of points $S'$ is not known, then it is not possible to choose among the infinite number of potential separating hyperplane which one is a correct one. And even if one is chosen, there is no way to tell if it is better or not than another.

In the context of machine learning, without an assessment criterion for a classification problem, it is not possible to learn a viable solution. Hence, we can not optimize the parameters. For binary classification, the Lemma 4.1 can be interpreted as: "The decision boundary between two classes can not be assessed nor learned if there is no access to data from both simultaneously".

**Lemma 4.2.** $\forall (S, S')$ *two bounded datasets not linearly separable. For any algorithm, it is impossible to assess a function $g(.)$ as a projection of $S$ and $S'$ into a space were they are linearly separable without access to $S'$ set.*

The proof is in appendix C, but in an insightful way, for any bounded set of points, there is an infinite number of projections of the initial set of point in a space where it could be linearly separable from another set of points. Then, If you don't know the second set of points $S'$ you can not choose among the infinite number of potential projections which one is a good one. And if you ever choose one, you have no way to tell if it is better or not than another. In the context of binary classification, the previous lemma can be interpreted as: "Two classes representation cannot be disentangled if there is no access to data from both simultaneously".

In those lemma, the concept of "not having access to" a certain dataset can both be applicable to not being able to sample data point from the distributions and to not have a model of the dataset. It can be generalized to not having access to any representative data distribution of a dataset.

## 4.2 Shortcomings in class-incremental tasks

We now prove that in incremental-class tasks, it is not possible to discriminate classes from different tasks using only a regularization based memory. The main point is that, to correctly learn to discriminate classes over different tasks the model needs access to both data distributions simultaneously.

In regularization methods, the memory only characterizes the model and the important parameters as explained in Section 3.2. This memorization gives insight on some past data characteristics but it is not a model of their distributions. If we take the cat vs dog example, a model that needs to discriminate white cats from black cats will learn to discriminate black features from white features. This "knowledge" can be saved in $\Omega$ but $\Omega$ will not save the full characteristics of a cat because the model never has to learn it. We bring then the following proposition:

**Proposition 4.3.** *While learning a sequence of disjoint classification tasks, if the memory $\Omega$ of the past tasks is only dependent on trained weights and learning criterion of previous task and does not model the past distribution, it is not possible for deep neural networks to learn new tasks without interference.*

*Proof.* The proof is organized in the following way: first, we present material necessary for the demonstration, then in a second part, we demonstrate that at any moment the classification task can be reduced to a binary classification task and in a third part we will show that we can not learn to solve this binary classification correctly.

**First part:** In the context of learning with a deep neural network, we can decompose the model into a non-linear feature extractor $g(\cdot)$ and an output layer to predict a class $y = \text{argmax}(\text{softmax}(A \cdot g(x) + b))$. With $A$ and $b$, respectively the matrix of projection and the bias of the linear layer. $\text{softmax}(.)$ is the output function that for a given class $i$ in a logits output $z$ gives $\text{softmax}(z_i) = \frac{e^{z_i}}{\sum_{j=0}^{N-1} e^{z_j}}$. The $\text{softmax}(.)$ function does not change the $\text{argmax}$ result and only regularize the output values and the gradient for later back propagation. We can thus remove it for our demonstration purposes.

The non-linear projection $g(.)$ should, therefore, disentangle classes and the linear output layer learns to predict the good class. The output layer allows for all classes $i$ to learn hyperplanes $A[:, i]$ with bias $b[i]$ such as: $\forall i \in [\![1, N]\!]$

$$\forall (\boldsymbol{x}, y) \in \mathbb{C}_t, \underset{i}{\text{argmax}}(A[:, i]h + b[i]) = y \tag{3}$$

with $h = g(x)$.

**Second part** For the sake of the demonstration, we would like to reduce the multi-classes classification problem into a binary classification problem. Hence, we can artificially split classes into two groups: classes from the past $\mathbb{Y}_{C_{t-1}}$ and current classes $\mathbb{Y}_{T_t}$.

We can then $\forall (\boldsymbol{x}, y) \in \mathbb{C}_t$ compute which class $\hat{y}_{C_{t-1}}$ upon the past classes $\mathbb{Y}_{C_{t-1}}$ is the most probable and compute which class $\hat{y}_{T_t}$ upon the current classes $\mathbb{Y}_{T_t}$ is the most probable.

$$\hat{y}_{C_{t-1}} = \underset{i \in \mathbb{Y}_{C_{t-1}}}{\operatorname{argmax}}(A[:,i]h + b[i]) \quad \text{and} \quad \hat{y}_{T_t} = \underset{i \in \mathbb{Y}_{T_t}}{\operatorname{argmax}}(A[:,i]h + b[i]) \tag{4}$$

Hence, the equation 3 can be rewritten into a binary operation:

$$\forall (\boldsymbol{x}, y) \in \mathbb{C}_t, \quad \underset{i \in \{\hat{y}_{C_{t-1}}, \hat{y}_{T_t}\}}{\operatorname{argmax}}(A[:,i]h + b[i]) = y \tag{5}$$

$$\begin{aligned} y &= \operatorname{argmax}(A[:,\hat{y}_{C_{t-1}}] \cdot h + b[\hat{y}_{C_{t-1}}], \; A[:,\hat{y}_{T_t}] \cdot h + b[\hat{y}_{T_t}]) \\ &= \operatorname{argmax}(0, (A[:,\hat{y}_{T_t}] - A[:,\hat{y}_{C_{t-1}}]) \cdot h + b[\hat{y}_{T_t}] - b[\hat{y}_{C_{t-1}}]) \end{aligned} \tag{6}$$

Equation 6 can directly be rewritten into the linear separability equation from definition 1. To make a proper decision, we should have $\forall (x, y) \in \mathbb{C}_t$, with $g(x) = h$ and $y = \hat{y}_{C_{t-1}}$ and $\forall (x', y') \in \mathbb{D}_t$, with $g(x') = h'$ and $y' = \hat{y}_{T_t}$.

$$(\boldsymbol{q} \cdot \boldsymbol{h} + q_0) \cdot (\boldsymbol{q} \cdot \boldsymbol{h'} + q_0) < 0 \tag{7}$$

Then, by identification, the classes $\hat{y}_{C_{t-1}}$ and $\hat{y}_{T_t}$ need to be separated by the hyperplane $\mathcal{Q}$ defined by a normal vector $\boldsymbol{q} = A[:,\hat{y}_{T_t}] - A[:,\hat{y}_{C_{t-1}}]$ and a position vector $q_0 = -(b[\hat{y}_{T_t}] - b[\hat{y}_{C_{t-1}}])$.

This binary classification description highlight that it is essential to be able to discriminate any class $\hat{y}_{C_{t-1}}$ from the past from any class $\hat{y}_{T_t}$ from the present for accurate predictions.

**Third part:** In this part, to prove proposition 4.3, we show that the model cannot learn the hyperplane $\mathcal{Q}$ from eq. 7.

To learn new tasks $T_t$ for $0 < t < N$, there are two different cases: *first* $g(\cdot)$ is already a good projection for $C_t$ tasks, i.e. classes are already disentangled in the embedded space. We assume that if classes are already disentangled, only the output layer has to be trained to solve $C_t$ tasks. *Secondly*, $g(\cdot)$ needs to be adapted, i.e. classes are not yet disentangled in the embedded space and new features need to be learned by $g(\cdot)$ to fix it. We refer as features, intrinsic characteristics of data that a model needs to detect to distinguish a class from another. We will show that it is not possible to learn to discriminate correctly the classes $\hat{y}_{C_{t-1}}$ from $\hat{y}_{T_t}$ from previous part.

*First case: Classes are disentangled*
Since we are in a regularization setting, at task $T_t$, we have access to $\Omega_{t-1}$ which contains classification information from previous tasks ($C_{t-1}$ tasks). However, by hypothesis, $\Omega_{t-1}$ does not model the data distribution from $C_{t-1}$ and therefore it does not model data distribution from $C_{t-1}$ classes.

Following from the second part of the proof, $\forall x \in C_t$ tasks, to make an accurate prediction, we need the right hyperplane $\mathcal{Q}$ that distinguish the most probable class from $C_{t-1}$, $\hat{y}_{C_{t-1}}$ and the most probable class from $T_t$, $\hat{y}_{T_t}$.

$\hat{y}_{C_{t-1}}$ and $\hat{y}_{T_t}$ classes images are a bounded set of points and $\hat{y}_{C_{t-1}}$ points are, by definition, not accessible, consequently following Lemma 4.1, it is impossible to assess a boundary between $\hat{y}_{T_t}$ and $\hat{y}_{C_{t-1}}$ even if by hypothesis this boundary exists. Therefore, we can not learn the hyperplane that discriminate $\hat{y}_{C_{t-1}}$ from $\hat{y}_{T_t}$ and ensure an accurate prediction.

*Second case: $g(\cdot)$ needs to be updated with new features.*
Let $\delta_{t-1}$ be the set features already learned by $g_{t-1}(\cdot)$ the feature extractor from previous task. $\Omega_{t-1}$ should keep $\delta_{t-1}$ unchanged while learning $T_t$. The goal is to make $\hat{y}_{C_{t-1}}$ and $\hat{y}_{T_t}$ linearly separable $\forall x \in \mathbb{C}_t$. Then, either $\delta_{t-1}$ already solve the problem and we are in *first case*, or a new set of features $\delta_t$ needs to be learned while learning $T_t$. In the second case, the set $\delta_t$ contains features to solve $T_t$, but features $\delta_{t-1:t}$ that distinguish classes from $T_{t-1}$ to classes from $T_t$ should also be learned. Then two cases raise, $\delta_{t-1:t} \not\subset \delta_t$ or $\delta_{t-1:t} \subset \delta_t$.

• if $\delta_{t-1:t} \not\subset \delta_t$, then supplementary features $\delta_{t-1:t}$ need to be learned. $\hat{y}_{C_{t-1}}$ and $\hat{y}_{T_t}$ classes images are a bounded set of points not linearly separable and since $\Omega_{t-1}$ does not give access to $C_{t-1}$ data

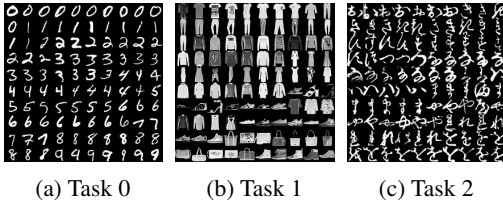

(a) Task 0      (b) Task 1      (c) Task 2

Figure 1: The three tasks of the MNIST-Fellowship dataset.

points, from Lemma 4.2 we can not assess a projection that put images from $\hat{y}_{T_t}$ and $\hat{y}_{C_{t-1}}$ into a linearly separable space, i.e. we can not learn the set of features $\delta_{t-1:t}$ to discriminate $\hat{y}_{C_{t-1}}$ images from $\hat{y}_{T_t}$ images and solve the continual problem.

• $\delta_{t-1:t} \subset \delta_t$ is possible, however, since data from $C_{t-1}$ are not available anymore, there is no way to project them in the new latent space with $\delta_t$ features. Therefore, without access of classes from both $C_{t-1}$ and $T_t$ tasks at time $t$ we can not identify $\delta_{t-1:t}$ features which are in $\delta_t$ features. It is also impossible to know if $\delta_{t-1:t} \subset \delta_t$. In other words, this case is not detectable and even if detected the features $\delta_{t-1:t}$ can not be used without data from $T_{t-1}$ (which is by definition prohibited).

In these two cases, there will be in any way conflict between losses leading to interference in the decision boundaries either because classes are not linearly separable or because a separation hyperplane cannot be found. In other words, the regularization methods can not discriminate classes from different tasks and they are then not suited to class-incremental settings.

$\square$

We can note that proposition 4.3, still holds if tasks are only partially disjoint, i.e. only some classes appear only once in the continual curriculum.

Indeed, in partially disjoint settings, several classes pairs are never in the same task. If we define two set of disjoint classes $Y$ and $Y'$, that will never be in the same task, the demonstration of proposition 4.3 can be applied on $Y$ and $Y'$. Then, classes $Y$ and $Y'$ will suffer from interference showing a shortcoming of regularization methods for this case too.

Therefore, if there is a class-incremental setting hidden into another setting, the regularization approach will not be able to solve it perfectly either. We could note that in many applications there are latent class-incremental problem to address in the learning curriculum. We mention some applications in Section 6.

A simple trick used in some regularization approaches to compensate their shortcomings is to use the task label for inferences, it gives a simple way to distinguish tasks from each other. However, it assumes the algorithms rely on a supervision signal for inferences. In the next section, we show that regularization shortcoming is easily highlighted with simple experiments.

## 5 EXPERIMENTS

To support the limitations presented earlier, we experiment with the "MNIST-Fellowship" dataset proposed in Lesort (2020). This dataset is composed of three datasets (Fig. 1): MNIST LeCun & Cortes (2010), Fashion-MNIST Xiao et al. (2017) and KMNIST Clanuwat et al. (2018), each composed of 10 classes, which should be learned sequentially one by one. We choose this dataset because it gathers three easy datasets for prototyping machine learning algorithms but solving those three quite different datasets is still harder than solving only one.

Our goal is to illustrate the limitation of regularization based methods in disjoint settings. In particular that they can not distinguish classes from different tasks. We would like also to show that the shortcoming happens both in the output layer and in the feature extractor. Thus, we propose three different settings with the *MNIST-Fellowship* dataset.

**1. Disjoint setting**: all tasks have different classes (i.e. from 0 to 9, 10 to 19 and 20 to 29, the output size is 30). This setting highlights shortcomings of regularization methods without test task labels.

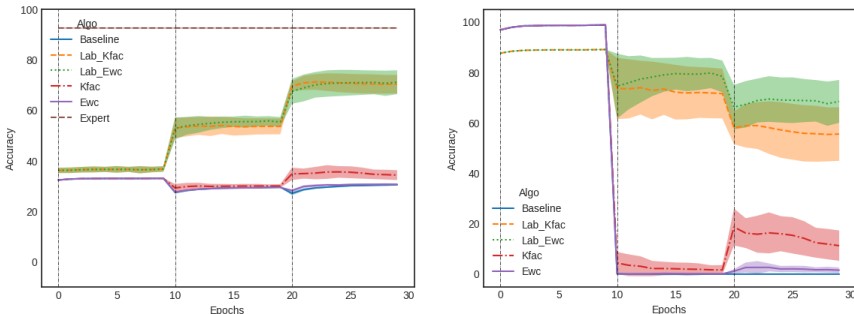

Figure 2: Experiment on disjoint classes without vs with test task label. Left, the mean accuracy of all 3 tasks, vertical lines are task transitions. Right, accuracy on the first task. Legends with 'Lab_' indicate experiments with task labels for test. The expert model is trained with i.i.d. data from all task and the baseline model is finetuned on each new task without any continual process.

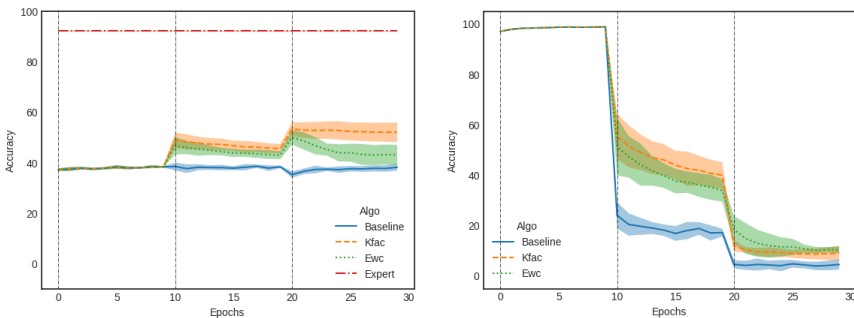

Figure 3: Experiments with joint classes. Left, mean accuracy of all 3 tasks, vertical lines are task transitions. Right, accuracy on the first task.

**2. Joint setting**: all tasks have the same classes ( i.e. from 0 to 9 for each task and the output size is 10) but different data. This scenario is designed as an instance incremental scenario Lesort et al. (2019c). This setting shows that they are interferences even when only the data instances change and not the class labels. Theoretically, this setting requires only the feature extractor to be adapted while learning a new task.

**3. Disjoint setting with test task label**: All tasks have different classes but at inference time, we know from which task a data-point is coming from. The output in this setting is a multi-head output with one head of size 10 for each task. This setting shows that regularization methods work when they have access to the test task labels.

With those settings, we present two experiments, the first one (Fig. 2) compares disjoint setting with and without a label for inferences. The goal is to bring to light that regularization fails in disjoint settings if the task label is not provided. Secondly, we experiment with the joint setting (Fig. 3), to show that even if the feature extractor only needs to be learned the approach still struggles to learn continually and forget.

We present EWC results with diagonal Fisher Matrix Kirkpatrick et al. (2017) and with Kronecker Factorization of the Fisher matrix Ritter et al. (2018). We add an expert model which learned all the datasets at once and a baseline model who learn without any memorization process. All models are trained with stochastic gradient descent with a learning rate of $0.01$ and a momentum of $0.9$. Even if continual learning does not support a-posteriori hyper-parameter selection, for fairness in comparison, the parameter lambda has been tuned. The best lambda upon $[0.1; 1; 2; 5; 10; 20; 100; 1000]$ is selected for each model. Then the model is trained on 5 different seeds.

The first experiment (Fig. 2), exhibits that *regularization* methods performances are significantly reduced when there is no test task label in the disjoint settings. The experiment also shows that without labels for inferences, the model forgets almost instantaneously the first task when switching to the

second one. Those results support the proposition 4.3. Indeed, the low performance of regularization methods without test task labels in disjoint settings illustrates the output layer shortcomings in continual learning (task separability problem example in appendix A).

In Experiment 2 (Fig. 3), since the classes are the same in all tasks, only the feature extractor needs to be learned continually. The low performance of the proposed models illustrates the shortcomings in the continual learning of the feature extractor (the latent features problem example in appendix A).

These two experiments show that learning continually with regularization is only efficient in the setting with task labels and maintains performance on task 0. The two other settings seem to either have interference in the output layer and in the feature extractor.

## 6 APPLICATIONS

In this section, we point out supplementary shortcomings of regularization in other types of learning situations, namely a classification task with one only class and multi-task continual reinforcement learning. We also use proposition 4.3 for the case of pre-trained models.

**- Learning from one class only**: A classification task with only one class might look foolish, however, in a sequence of tasks with varying number of classes, it makes more sense and it seems evident that a CL algorithm should be able to handle this situation. Nevertheless, a classification neural network needs at least two classes to learn discriminative parameters. Hence, in a one-class task, the model learns no useful parameters, a regularization term can then a fortiori not protect any knowledge. As noted in Lesort et al. (2019b), the regularization method is not suited for such setting. It is worth noting that in a real-life settings it is mandatory to be able to learn only one concept at a time.

**- Multi-task Continual Reinforcement Learning:** Results from section 4.2 can also be generalized to continual multi-tasks reinforcement learning settings Traoré et al. (2019). In this setting, a model has to learn several policies sequentially to solve different tasks. At test time with no task label, the model needs to both be able to run the policies correctly but also to infer which policy to run. However, since policies are learned separately inferring which one to run is equivalent to a class incremental task. Therefore, following proposition 4.3, the regularization based method will not be able to learn the implicit classification correctly. Hence, in continual multi-tasks RL a regularization method alone will fail if task label is not enabled at test time.

**- Using pre-trained models for continual learning:** We showed in Section 4.2 that, in a class incremental classification scenario, regularization methods are not sufficient to learn continually. In the case of a pre-trained classification model on $N$ classes that we want to train on new classes without forgetting, if the training data are not available for some reasons, then we don't even have a *regularization term* $\Omega$ to protect some. Following the proposition 4.3 and a fortiori without the regularization term, the model will forget past knowledge while learning new classes. Using pre-trained models can be useful to converge faster to a new task solution but it will undoubtedly forget what it has learn previously.

## 7 DISCUSSION AND CONCLUSION

Regularization is a widespread method for continual learning. However, we prove that for class-incremental classification, no regularization method can continually learn properly the decision boundaries. At test time, this shortcoming makes them dependant on the task label for prediction.

The class-incremental scenarios are benchmarks measuring the ability of algorithms to learn sequentially different classes. However, being unable to deal with this setting implies that in a more complex learning environment, all sub-tasks assimilable to a class-incremental task will be failed.

It is fundamental for continual learning to produce algorithms that can autonomously learn and be deployed. Algorithms that rely on the test task label are not autonomous and therefore should be avoided. This paper shows that in any setting where a class-incremental setting may be hidden, regularization methods alone should be avoided. A fortiori, in continual learning the future is unknown, therefore future tasks might add new classes or not. Then, in order to deal with new tasks whatever their nature, regularization methods alone should always be avoided on continual learning applications.

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

## A PRACTICAL EXAMPLES

To illustrate the proposition from section 4.2, we present two insightful examples of regularization limitations.

**- The Task Separability Problem:**

In the first case of proposition 4.3 proof, we already have a perfect feature extractor. Classes are already linearly separable and only the output layer needs to be learned continually.

If we have only two classes in the first task, the model will learn one hyperplane $\mathcal{Q}_0$ separating the instances of these two classes (See Figure 4). For the second task, we have two new classes and a regularization protecting $\mathcal{Q}_0$. Then, we can learn a hyperplane $\mathcal{Q}_1$ that separates our two new classes. In the end, we have learned the hyperplanes $\mathcal{Q}_0$ and $\mathcal{Q}_1$ to distinguish classes from $T_0$ and classes from $T_1$. But none of those hyperplanes helps to discriminate $T_0$ classes from $T_1$ classes, as illustrated Figure 4. This will lead to error in the neural networks predictions.

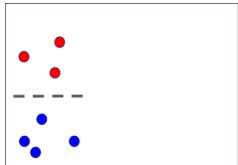 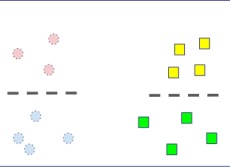

Figure 4: Simple case of continual learning classification in a multi-task setting. Left, the task T0: learning a hyperplane splitting two classes (red and blue dots). Right, the task T1: learning a line splitting two classes (yellow and green squares) while remembering $T_0$ models without remembering $T_0$ data (pale red and blue dots).

**- The Latent Features Problem:**

In the second case of Proposition 4.3 proof, the feature extractor needs to be updated to learn new features extractors.

If we have only two classes in the first task, the model will learn to separate classes instances into two groups with the features extractor $g_0$ and one hyper-plan $\mathcal{Q}_0$ separating the two classes instances (See Figure 5).

For the second task, we have two new classes and a regularization protecting $\mathcal{Q}_0$ and $g_0$. Then, we can learn a features extractor $g_1$ to disentangle new class instances in the latent space and a hyperplane $\mathcal{Q}_1$ that separates them. In the end, we can disentangle classes from $T_0$ and classes from $T_1$ and we have two hyperplanes $\mathcal{Q}_0$ and $\mathcal{Q}_1$ to distinguish classes from $T_0$ and classes from $T_1$. But we can not disentangle $T_0$ classes from $T_1$ classes and none of the learned hyperplanes helps to discriminate $T_0$ classes from $T_1$ classes (See Fig. 6). It leads to errors in the neural network predictions. At test time, it will not be possible for the model to discriminate between classes correctly.

However, with the task label for inferences, we could potentially perfectly use $g_0$, $g_1$, $\mathcal{Q}_0$ and $\mathcal{Q}_1$ to make correct predictions. Nevertheless, assuming that the task label is available for prediction is a strong assumption in continual learning and involves a need of supervision at test time.

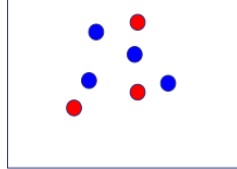 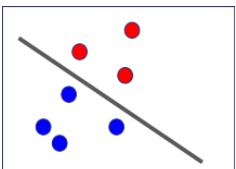

Figure 5: $\mathbb{D}_0$ feature space before learning $T_0$ (Left), $\mathbb{D}_0$ feature space after learning $T_0$ with a possible decision boundary (Right). Data points are shown by blue and red dots. The line (right part) is the model learned to separate data into the feature space.

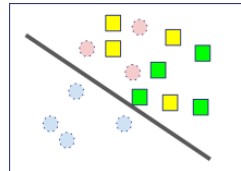 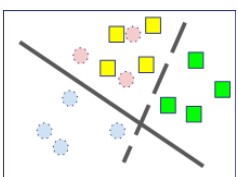

Figure 6: Case of representation overlapping while continual learning classification in a multi-task setting. At task $T_1$, feature space of $\mathbb{D}_1$ before learning $T_1$ (Left), Feature space of $\mathbb{D}_1$ after learning $T_1$ with a possible decision boundary (Right). New data are plotted as yellow and green squares and old data that are not available anymore to learn are shown with pale red and blue dots.

## B  PROOF OF LEMMA 4.1

**lemma.** $\forall (S, S')$ *bounded set of discrete points in $R^n$ and linearly separable by a hyperplane $\mathcal{Q}$. For any algorithm, it is impossible to assess $\mathcal{Q}$ as a separation hyperplane without $S'$ set.*

*Proof.* Let $S$ and $S'$ be two bounded and linearly separable set of discrete points in $R^n$. Let $\mathcal{Q}$ be a potential linear separation between $S$ and $S'$. The hyperplane $\mathcal{Q}$ can not be assessed as a linear separation between $S$ and $S'$ if there exists at least one hyperplane indistinguishable from $\mathcal{Q}$ and which is not a separation boundary between $S$ and $S'$. Let $\mathcal{P}$ be a hyperplane, defined as a normal vector $\boldsymbol{p}$ and position vector $p_0$, is a separation boundary between $S$ and $S'$ if all the point of $S$ are on one side of $\mathcal{P}$ and all point of $S'$ are on the other side. It can be formalized as follows:

$\forall \boldsymbol{x} \in S$ & $\forall \boldsymbol{x'} \in S'$:

$$(\boldsymbol{p} \cdot \boldsymbol{x} + p_0) \cdot (\boldsymbol{p} \cdot \boldsymbol{x'} + p_0) < 0 \tag{8}$$

Where $< \cdot >$ is the scalar product.

Without the access of $S'$, eq. 8 can not be evaluated. However, we can evaluate it, if all the point of $S$ are on the same side of $\mathcal{P}$

Eq. 8, verify that $S$ and $S'$ are each entirely on different side of the $\mathcal{P}$. By definition if all the point of $S$ are above $\mathcal{P}$ then:

$\forall \boldsymbol{x} \in S$

$$(\boldsymbol{p} \cdot \boldsymbol{x} + p_0) > 0 \tag{9}$$

If all the point are under $\mathcal{P}$ then:

$$(\boldsymbol{p} \cdot \boldsymbol{x} + p_0) < 0 \tag{10}$$

And if neither eq. 9 nor eq. 10 are verified then all the points of $S$ are not on the same side of $\mathcal{P}$.

Finally, we can merge both 9 and eq. 10 and verify only:

$\forall \boldsymbol{x} \in S$

$$sign(\boldsymbol{p} \cdot \boldsymbol{x} + p_0) = constant \tag{11}$$

Where $sign(.)$ is the function which returns the sign of any real value.

The Lemma 4.1 is proven if $\exists\,\mathcal{P}$ such as eq. 11 is true but not eq. 8, because $\mathcal{P}$ would not be a linear separation of $S$ and $S'$ and would not be distinguishable from $\mathcal{Q}$ without access to $S'$.

Now, we will build an hyper-plan $P$ that is unquestionably respect eq. 11 and not eq. 8.

We know that $S$ is bounded, then it has both upper and lower bounds in all the direction of $R^n$. If eq. 11 is respected, then $\mathcal{Q}$ is a bound of $S$ in the direction of its normal vector $\boldsymbol{q}$. If we move $\mathcal{Q}$ along the direction of $\boldsymbol{q}$ (i.e. if we change $q_0$ the position vector), we can find at least one other plane $\mathcal{P}$ respecting eq. 11: the opposing bound of $S$ along the direction $\boldsymbol{q}$.

Since, $\mathcal{P}$ and $\mathcal{Q}$ are two opposing bounds of $S$ in the same direction $\boldsymbol{q}$, then:

$\forall x \in S$

$$sign(\boldsymbol{p} \cdot \boldsymbol{x} + p_0) \neq sign(\boldsymbol{q} \cdot \boldsymbol{x} + q_0) \tag{12}$$

If $\mathcal{Q}$ is a lowerbound of $S$ in the direction $\boldsymbol{q}$ and an upperbound of $S'$ in the same direction then, a lowerbound of $S'$ in the direction $\boldsymbol{q}$ is a lowerbound of $S$ in the same direction and an upperbound of $S$ in the direction $\boldsymbol{q}$ is an upperbound of $S'$ in the same direction. (We leave the demonstration to the reader).

Therefore, $\mathcal{Q}$ and $\mathcal{P}$ are both upperbounds or both lowerbounds of $S'$ in the direction of $\boldsymbol{q}$. $\forall x' \in S'$:

$$sign(\boldsymbol{p} \cdot \boldsymbol{x'} + p_0) = sign(\boldsymbol{q} \cdot \boldsymbol{x'} + q_0) \tag{13}$$

Then with 12 and eq. 13:

$$(\boldsymbol{p} \cdot \boldsymbol{x} + b) \cdot (\boldsymbol{p} \cdot \boldsymbol{x'} + b) > 0 \tag{14}$$

Consequently, from eq 11 and eq 14, $\exists$ a hyperplane $\mathcal{P}$ which respects eq. 11 and not eq 8, $\mathcal{P}$ is indistinguishable from $\mathcal{Q}$ and is not a separation boundary between $S$ and $S'$.

$\square$

## C   PROOF OF LEMMA 4.2

**lemma.** $\forall (S, S')$ *two bounded datasets not linearly separable. For any algorithm, it is impossible to assess a function $g(.)$ as a projection of $S$ and $S'$ into a space were they are linearly separable without $S'$ set.*

*Proof.* $g(.)$ is a projection of $S$ and $S'$ into a space where they are linearly separable means:

$\forall \boldsymbol{x} \in S$ & $\forall \boldsymbol{x'} \in S'$, then $g(x)$ and $g(x')$ respect eq. 8.

Without access to $S'$ this condition can not be verified. However, we can verify eq. 11 with $g(x)$.

The Lemma 4.2 is proven if $\forall \boldsymbol{x} \in S$ & $\forall \boldsymbol{x'} \in S'$, $\exists$ a projection $f$, that respect eq. 11 with $f(x)$ but not eq. 8 with $f(x)$ and $f(x')$, because then $f$ and $g$ are indistinguishable without access to $S'$.

Let $f$ be the identity function, $\forall z \in \mathbb{R}\ f(z) = z$. We define $S_f$ and $S'_f$, the set of point $S$ and $S'$ after projection by $f$. Since $f$ is the identity function, $S$ and $S'$ are respectively identical to $S_f$ and $S'_f$. Since $S$ is bounded, $S_f$ is also bounded. Hence there exists a hyperplane $\mathcal{P}$ that verify eq. 11 with $f(x)\ \forall x \in S$. By hypothesis, $S$ and $S'$ are not linearly separable so $S_f$ and $S'_f$ is also not linearly separable. Then $\nexists!$ hyperplane $\mathcal{P}$ which respect eq. 8 with $f(x)$ and $f(x')$.

Thus, $f$ exists and therefore it is impossible to assess any function as a projection of $S$ and $S'$ into a space were they are linearly separable without $S'$ set.

$\square$

## D   REGULARIZATION METHODS

To illustrate the previous section, we present several famous regularization methods in our formalism.

**- Elastic Weight Consolidation** (EWC) Kirkpatrick et al. (2017) is one of the most famous regularization approaches for continual learning. The loss augmented with a regularization term is at task $t$:

$$\ell_{\mathbb{C}_t}(\boldsymbol{\theta}) = \ell_{\mathbb{D}_t}(f(x; \boldsymbol{\theta}), y) + \frac{\lambda}{2} * F_{t-1}(\boldsymbol{\theta}^*_{t-1} - \boldsymbol{\theta})^2 \tag{15}$$

with $(.)^2$ the element-wise square function.

We can then by identification, extract our function $\Omega_t(\boldsymbol{\theta}^*, \ell_D, \boldsymbol{\theta})$

$$\Omega_t(\boldsymbol{\theta}^*, \ell_{\mathbb{C}_{t-1}}, \boldsymbol{\theta}) = \frac{1}{2} * F_{t-1}(\boldsymbol{\theta}^*_{t-1} - \boldsymbol{\theta})^2 \tag{16}$$

$F_t$ is a tensor of size $card(\boldsymbol{\theta})^2$, specific to task $t$, characterizing the importance of each parameter $\theta_k$. $F_t$ is computed at the end of each task and will protect important parameters to learn without forgetting. In EWC, the $F_t$ tensor is implemented as a diagonal approximation of the Fisher Information Matrix:

$$F_t = \mathbb{E}_{(\boldsymbol{x},y)\in\mathbb{D}_t}\left[\left(\frac{\partial log\ p(\hat{y})}{\partial\boldsymbol{\theta}}\right)^2\right] \tag{17}$$

where $\hat{y} \sim P(f(\boldsymbol{x};\boldsymbol{\theta}))$. The diagonal approximation allows to save only $card(\boldsymbol{\theta})$ values in $F_t$.

**- K-FAC Fisher approximation** Ritter et al. (2018) is very similar to EWC but approximates the Fisher matrices with a Kronecker factorization (K-FAC) Martens & Grosse (2015) to improve the expressiveness of the posterior over the diagonal approximation. However, the Kronecker factorization saves more values than the diagonal approximation.

**- Incremental Moment Matching** (IMM) Lee et al. (2017) proposes two regularization approaches for continual learning which differ in the computation of the mean $\theta_{0:t}$ and the variance $\sigma_{0:t}$ of the parameters on all tasks.

The idea is to regularize parameters such that the moments of their posterior distributions are matched in an incremental way. It means that each parameter is approximated as a normal distribution and their mean or standard deviation should match from one task to another. This regularization, on the parameters' low-order moments, helps to protect the model from forgetting.

*- Mean based Incremental Moment Matching (mean-IMM)*

$$\theta_{0:t} = \sum_{i=0}^{t}\alpha_i\boldsymbol{\theta}_i^* \quad \text{and} \quad \sigma_{0:t} = \sum_{i=0}^{t}\alpha_i(\sigma_i + (\boldsymbol{\theta}_i^* - \boldsymbol{\theta}_{0:t})^2) \tag{18}$$

$\alpha_i$ are importance hyper-parameters to balance past task weight into the loss function. They sum up to one.

*- Mode based Incremental Moment Matching (mode-IMM)*

$$\boldsymbol{\theta}_{0:t} = \sigma_{0:t}\cdot\sum_{i=0}^{t}(\alpha_i\sigma_i^{-1}\boldsymbol{\theta}_i^*) \quad \text{and} \quad \sigma_{0:t} = (\sum_{i=0}^{t}\alpha_i\sigma_i^{-1})^{-1} \tag{19}$$

$\sigma_i$ is computed as the Fisher matrix (eq. 17) at task $i$.

Then at task $t$, with $\theta_{0:t-1}$ and $\sigma_{0:t-1}$ we can compute:

$$\Omega_t(\boldsymbol{\theta}^*,\ell_{\mathbb{C}_{t-1}},\boldsymbol{\theta}) = \frac{1}{2}\sigma_{0:t-1}(\boldsymbol{\theta}_{0:t-1} - \boldsymbol{\theta})^2 \tag{20}$$

**- Synaptic Intelligence:** (SI) Zenke et al. (2017) The original idea is to imitate synapse biological activity. Therefore, each synapse accumulates task relevant information over time, and exploits this information to rapidly store new memories without forgetting old ones. In this approach, we can identify $\Omega_t$ as:

$$\Omega_t(\boldsymbol{\theta}^*,\ell_{\mathbb{C}_{t-1}},\boldsymbol{\theta}) = M_t(\boldsymbol{\theta}_{t-1}^* - \boldsymbol{\theta})^2 \tag{21}$$

$M_t$ is a tensor of size $card(\boldsymbol{\theta})$ specific to task $t$ characterizing the importance of each parameter $\theta_k$ over the all past tasks such as:

$$M_t = \sum_{0<i<t}\frac{\boldsymbol{m_i}}{\boldsymbol{\Delta}_i^2 + \xi} \tag{22}$$

$M_t$ is the sum over $m_i$ which characterizes the importance of each parameter on task $i$, with $\Delta_i = \boldsymbol{\theta}_i^* - \boldsymbol{\theta}_{i-1}^*$. $\xi$ is a supplementary parameter to avoid null discriminator.

$$m_i = \int_{T_{i-1}}^{T_i}\nabla_{\boldsymbol{\theta}}\delta_{\boldsymbol{\theta}}(t)dt \tag{23}$$

With $\delta_{\boldsymbol{\theta}}(t)$ the parameter update at time step $t$.

# E IMPLEMENTATION DETAILS

## E.1 DATA PREPROCESSING

All data points were preprocessed to be between 0 and 1 by a division by 255.

## E.2 DATASETS SPLITTING

For all datasets used, we selected randomly 20% of the train set for validation and used the original split test/train of the datasets for the test sets and the train sets.

## E.3 COMPUTING INFRASTRUCTURE

The experiments were run with a GPU GeForce GTX 1080 Ti with a CPU Intel Core i7-7700K @ 4.2 GHZ x 8.

## E.4 NUMBER OF EVALUATION RUNS

A single evaluation run have been executed after trainning.

# F MODEL ARCHITECTURE

Table 1: Model architecture, convolution have 5*5 kernel size, maxpool have 2*2 kernel size. Parameters not mentioned are default parameters in Pytorch library Paszke et al. (2019) (in torch.nn). BS is for batch size, which is 64. All layers are initialized with Xavier init method Glorot & Bengio (2010).

| Layer Name | Layer Type | Input Size | Output Size |
|---|---|---|---|
| Conv1 | ReLu(MaxPool2d(Conv2d(input))) | BS*1*28*28 | BS*10*12*12 |
| Conv2 | ReLu(MaxPool2d(Conv2d(input))) | BS*10*14*14 | BS*20*4*4 |
| Linear1 | ReLu(Linear(input)) | BS*320 | BS*50 |
| Linear2 | functional.log_softmax(Linear(input)) | BS*50 | BS*10 |

