# OpenReview forum: "Regularization Shortcomings for Continual Learning"
_ICLR.cc/2021/Conference — Reject_

### Official Review · AnonReviewer1 · 2020-10-24
**What could be an interesting contribution lacks the formal language to prove its claims**

**Rating:** 4
**Confidence:** 4

**Review:**

TL;DR: The paper presents an intuition as to why continual learning with regularization will be unable to discern subsequent tasks. While I believe the intuitions to be correct, the formal laguage used to present this intuition is not sufficiently rigorous to warrant publication.

The paper sets out to build on the surprisingly sparse literature on the theory of continual learning. The key point the paper wishes to make is that regularization-based Continual Learning approaches are unable to discern tasks without additional information such as task labels at training time. This is attempted in the form of three propositions.

First of, I should say that I believe the authors are on the right track & have the correct intuitions: I believe that the claim they want to make is correct, and I believe that classification is a promising setting to prove their claim via separating hyperplanes. In other words, I want to like the paper but its current state does not allow me to recommend its publication. Let me also be transparent: I think this paper needs more than a bandaid, and will in my opinion not be publishable without a ‘major revision’/‘revise and resubmit’ order of magnitude in treatment (to use journal jargon). In essence, my problem is that I do not think the current version of the paper manages to deliver a convincing rigorously mathematical proof of the developed intuitions. Since this is what the paper sets out to do, I thus recommend that its current form not be published.

The main hindrance throughout is a serious lack in precision. This begins with the definitions: For instance, Definition 2 defines 'interferences' as follows:
"In machine learning, interferences are conflicts between two (or more) objective functions leading to prediction errors. There are interferences when optimizing one objective function degrades the optimization of, at least, another one."
While this definition captures the intuition of interference, it is not precise enough (e.g., which objective functions, what kind of prediction error, … ?) to be used within a mathematical result as is done in Proposition 4.3, which states that
"While learning a sequence of disjoint classification tasks, if the memory $\Omega$ of the past tasks is only dependent on trained weights and learning criterion of previous task and does not model the past distribution, it is not possible for deep neural networks to learn new tasks without interference."
For example, it is unclear which 'prediction error' (which is part of the interference definition from Def. 2) or which objective functions (again, they are part of the definition 2) is/are the relevant one/ones for Proposition 4.3. While both answer can be intuited to a certain degree if the reader is eager enough, a rigorous result requires the reader to exactly know all relevant objects in question. What "model[ling] the past distribution" is also not precisely defined, but used to state this result.

This kind of imprecision plagues the paper throughout. E.g., while Lemma 4.1 refers to $(S, S')$ as "bounded set of discrete points in R^n" (which is very clear), Lemma 4.2 suddenly calls the same tuple "two bounded datasets" without previously defining datasets as bounded sets of discrete points in R^n. While this particular imprecision can be intuited, this sort of imprecision does not help me trusting the result.

Similar problems plague the proof of Proposition 4.3 — my criticism is no that it is false. Instead, my criticism is that it is written in such vague terms that it becomes impossible to assess if it is correct or incorrect.  In fact, I believe that most of the intuitions I can follow are correct. But this is the problem: the proofs are written in an imprecise way, forcing me to believe rather than convince myself: As the definitions and wording throughout the paper, the statements in the proof lack rigour and are much more informal than what would be required for a watertight proof.  As I understand it, the main thrust of the proof is the following idea: It is possible to rewrite the binary classification problem as the problem of finding a hyperplane between classes. One can further rewrite the continual learning problem as the attempt of finding a hyperplane between a new class and previous classes. By Lemmas 4.1/4.2, this tells us that the problem is not solvable unless we store everything we observed previously in the regulariser. I actually believe that this idea is very elegant, simple, and could be shown rigorously. But I don’t believe the current paper does that. I think comparing the current paper’s approach with the related literature [1,2,3] will benefit the authors, and allow them to clearly define the relevant objects of interest & rigorously prove their interesting result.

Regarding the lemmas, I will also note the following: While both results are clearly correct (in the sense that without the second set of points, the separating hyperplane problem is not even defined), the purpose of Lemmas 4.1/4.2 is doubtful. It is fairly clear that you will not be able to find the separating hyperplane between two collections of points if you are given only one of these collections. Accordingly, the proof of these lemmas feels  like an attempt at proving that you won't know what real number x+y is equal to if you are only given x=3, but y remains unknown. What I am trying to say is that the very problem of finding a separating hyperplane (or of finding the real number x+y) is not defined if one of the two collection of points (or y) is not given.

Though the following is less important than my main points, I also would have enjoyed a more thorough review of the (very sparse) related literature. Since there are very few papers discussing the theory of continual learning, this part should be an easy fix. To the best of my knowledge, there are essentially 2, maybe 3 relevant papers.
[1] https://arxiv.org/abs/2006.11942 [unpublished, deriving generalization bounds for Continual Learning]
[2] https://arxiv.org/abs/2006.05188 [ICML 2020, probably most related to the current paper as they basically show a different ‘impossibility’ result than Proposition 4.3 which I think holds more generally]
[3] https://arxiv.org/abs/1610.08628 [JMLR 2017, probably least related but still worth discussing, derives regret bounds]

---

### Official Review · AnonReviewer3 · 2020-10-26
**Interesting conclusion but insufficient experimental analysis**

**Rating:** 5
**Confidence:** 4

**Review:**

The paper presents an interesting conclusion that kinds of regularizations cannot learn to discriminate classes from different learning tasks. Both theoretical and experimental analysis is provided. Two regularization methods are tested.

Pro:
1.	The conclusion is interesting and it reveals that regularizations fail when the task label isn’t given. This conclusion questions those proposed regularizations for continual learning.
2.	Theoretical analysis of the shortcomings of regularizations in continual learning is presented. Besides, the paper also pointed out supplementary shortcomings of regularization in other types of learning situations.

Cons:
1.	The related work is not sufficient. It lacks the introduction and discussion of popular regularization methods in continual learning.
2.	The experiments are implemented on 3 tiny datasets (MNIST, Fashion-MNIST, KMNIST). It’s not sufficient to show the importance of the proposed conclusion. The popular CIFAR100 dataset should be tested.
3.	Only two regularization methods are tested. The popular knowledge distillation based regularization [a] is not tested.
4.	Although regularizations cannot learn to discriminate classes from different tasks, they can avoid the bias to new training data, which is also important for improving CL performance. The paper would be stronger if the analysis why regularizations can improve CL performance is presented.
5.	Can you explain why there is a sharp promotion/decrease of accuracy at the final steps in each learning task (Figure 2, 3)?

[a] Learning without Forgetting, Li et al, 2018.

---

### Official Review · AnonReviewer4 · 2020-10-28
**Review for Regularization Shortcomings for Continual Learning**

**Rating:** 5
**Confidence:** 2

**Review:**

This paper presents a theoretical analysis of regularization based approaches to the problem of continually learning a sequence of tasks. The point of the paper is to demonstrate shortcomings of these kinds of approaches, in the context of class-incremental learning where classes are observed once and one after another. The authors argue that these kinds of methods require task labels at test time to correctly distinguish classes from different tasks.

I would like to mention as strengths of the paper:
-	The paper is a valuable attempt at examining properties of existing approaches to the problem of continual learning. I appreciate the rigorous approach to the evaluation of the shortcomings of these kinds of methods.
- The paper is in general well-written and easy to follow. There are no major errors or typos.

The weaknesses I see in this paper are:
-	Although there is a clear and formal explanation of why it is not possible to discriminate among classes from different task when there is no access to data from those previous classes, I am not fully convinced that the set of parameters kept from previous classes, and used in regularization-based approaches, do not represent to some extent this data. In particular, there is no clear argument for the claim on page 5: “However, by hypothesis, \omega_{t-1} does not model the data distribution from C_{t-1} and therefore it does not model data distribution from C_{t-1}  classes.”. I would like to see some discussion regarding how fairly a set of parameters \theta_{t-1} would represent the S’ set.
-	In terms of the experiments, I consider the number of tasks quite limited. To be convinced I would like to see several tasks (at least 10) and sequential results in terms of tasks learned rather than epochs.

Questions for authors:
- Please address my comments on the weaknesses above.

---

### Official Review · AnonReviewer2 · 2020-10-28
**Interesting hypothesis, but the paper doesn't validate the claims being made**

**Rating:** 3
**Confidence:** 5

**Review:**

This paper argues that regularization-based approaches to continual learning fail to distinguish between classes from different tasks in the class-incremental setting (without relying on “task labels” to simplify the problem). The paper uses a theoretical argument showing it is not possible to assess the discriminability between two classes from different tasks without access to the data (or a model of the data distribution), and also demonstrates a difference in performance when conditioning on task labels.

I think the thesis of the paper is plausible, but unfortunately, I don’t think this is demonstrated in the submission in its current form, and think there are a few flaws in the argument that need to be addressed.

I do think this is an interesting and important line of inquiry, and could lead to valuable insights (eg. proving comprehensively that regularization-based approaches need to do a better job of capturing the data distribution, or showing they can be improved by doing so).

I encourage the authors to address the comments below in lieu of this.

Main comments:
1) The argument relies heavily on the assumption that regularization-based approaches to supervised continual learning do not model the data distribution, eg. “However by hypothesis, [the regularization-based model] does not model the data distribution ….”
First, this seems to conflate regularization-based methods and discriminative models, as regularization-based methods can be applied to generative modelling tasks (eg. VCL [1]), with a classifier used on learned representations - this scenario is not considered here. Second, there is work showing that discriminative models have some generative capabilities (eg. [2]), and it may be possible to extract this information to better model the data distribution and understand where previously learned decision boundaries are valid.
2) The theoretical reasoning essentially demonstrates the simple intuition that we cannot assess whether the samples from two classes are linearly separable without having access to both sets of data or a model of the data from the class from the previous task. This insight makes sense, but it’s not clear that it is non-trivial or important: it says nothing about how well (and under what conditions) previously learned decision boundaries may generalise/transfer to a new task, and ignores my point (1) above that discriminative models do carry information about the data distribution.
3) The empirical results primarily show that performance with some regularization methods is much worse when not using the task labels, in different situations (eg. with disjoint/joint classes in each task). This is known from previous work, since regularization-based methods get much poorer performance in class-incremental learning (ie. single head, no task label) compared to task-incremental learning (multi-head, task label given) - see for example [3][4].
Unfortunately, these results alone cannot attribute the poor performance to the reasons argued in the paper (ie. the inability to capture previous data distributions, leading to the inability to discriminate classes across tasks), and it would be nice to see experiments that actually measure whether the poor performance is due to the arguments made. The paper also makes a broader claim about regularization-based methods being limited, but only evaluates EWC and K-FAC. Other regularization-based approaches like VCL (without coreset)[1] and BGD [5] could be explored too.
4) The writing is quite difficult to follow at times: I’d recommend another thorough proofread to address grammar /spelling and clarity issues. See a list of some examples below, but there are many others.
5) There are a number of papers in continual learning that have been missed - both older regularization-based approaches, and some newer ones (see refs below)

Some writing issues:
- “inferences” should be “inference” in most places in the text. . eg. “for inference”, “during inference”, etc.
- Most citations seem to not have parentheses, so they blend into the text. (Eg. “for continual learning Kirkpatrick et al. (2017); …” on page 1). Please use \citep instead of \citet when the citation doesn’t flow from the text.
- Section 3.2: “The regularization term omega act…” should be acts.
- Also Section 3.2: “a matrix pondering weights importance” - not sure what ‘pondering’ should mean here.
- Definition 1, “...separating two dataset…”, should be “datasets”
- I think “disentangled” should be “discriminable” in the Third Part section of Section 4.2.

References:
[1] Nguyen, Cuong V., et al. "Variational continual learning." arXiv preprint arXiv:1710.10628 (2017).
[2] Grathwohl, Will, et al. "Your classifier is secretly an energy based model and you should treat it like one." arXiv preprint arXiv:1912.03263 (2019).
[3] Hsu, Yen-Chang, et al. "Re-evaluating continual learning scenarios: A categorization and case for strong baselines." arXiv preprint arXiv:1810.12488 (2018).
[4] van de Ven, Gido M., and Andreas S. Tolias. "Three scenarios for continual learning." arXiv preprint arXiv:1904.07734 (2019).
[5] Zeno, Chen, et al. "Task agnostic continual learning using online variational bayes." arXiv preprint arXiv:1803.10123 (2018).
[6] Aljundi, Rahaf, et al. "Memory aware synapses: Learning what (not) to forget." Proceedings of the European Conference on Computer Vision (ECCV). 2018.
[7] Aljundi, R., Lin, M., Goujaud, B., & Bengio, Y. (2019). Gradient based sample selection for online continual learning. In Advances in Neural Information Processing Systems (pp. 11816-11825).
[8] Rao, D., Visin, F., Rusu, A., Pascanu, R., Teh, Y. W., & Hadsell, R. (2019). Continual unsupervised representation learning. In Advances in Neural Information Processing Systems (pp. 7647-7657).
[9] Chaudhry, Arslan, et al. "Using hindsight to anchor past knowledge in continual learning." arXiv preprint arXiv:2002.08165 (2020).

---

### Decision · Program_Chairs · 2021-01-07
**Final Decision**

**Decision:**

Reject

**Comment:**

All four reviewers recommend rejecting the paper. However there is agreement that this is an interesting line of research, and the AC agrees. Reviewers provided extensive and well educated feedback. The authors did not respond to the raised concerns.